# Altered Secretome of Diabetic Monocytes Could Negatively Influence Fracture Healing—An In Vitro Study

**DOI:** 10.3390/ijms22179212

**Published:** 2021-08-26

**Authors:** Caren Linnemann, Lorena Savini, Mika F. Rollmann, Tina Histing, Andreas K. Nussler, Sabrina Ehnert

**Affiliations:** Siegfried Weller Research Institute, Department of Trauma and Reconstructive Surgery, BG Trauma Center Tübingen, University of Tübingen, Schnarrenbergstr. 95, D-72076 Tübingen, Germany; caren.linnemann@student.uni-tuebingen.de (C.L.); lorena.savini@gmx.de (L.S.); mrollmann@bgu-tuebingen.de (M.F.R.); thisting@bgu-tuebingen.de (T.H.); andreas.nuessler@gmail.com (A.K.N.)

**Keywords:** diabetes mellitus, fracture, mononuclear cells, osteoprogenitor cells, migration, chemokines, ERK1/2

## Abstract

Diabetes mellitus is a main risk factor for delayed fracture healing and fracture non-unions. Successful fracture healing requires stimuli from different immune cells, known to be affected in diabetics. Especially, application of mononuclear cells has been proposed to promote wound and fracture healing. Thus, aim was to investigate the effect of pre-/diabetic conditions on mononuclear cell functions essential to promote osteoprogenitor cell function. We here show that pre-/diabetic conditions suppress the expression of chemokines, e.g., *CCL2* and *CCL8* in osteoprogenitor cells. The associated MCP-1 and MCP-2 were significantly reduced in serum of diabetics. Both MCPs chemoattract mononuclear THP-1 cells. Migration of these cells is suppressed under hyperglycemic conditions, proposing that less mononuclear cells invade the site of fracture in diabetics. Further, we show that the composition of cytokines secreted by mononuclear cells strongly differ between diabetics and controls. Similar is seen in THP-1 cells cultured under hyperinsulinemia or hyperglycemia. The altered secretome reduces the positive effect of the THP-1 cell conditioned medium on migration of osteoprogenitor cells. In summary, our data support that factors secreted by mononuclear cells may support fracture healing by promoting migration of osteoprogenitor cells but suggest that this effect might be reduced in diabetics.

## 1. Introduction

With approximately 440 million patients reported in 2019, type 2 diabetes mellitus (T2DM) is the metabolic disorder with the highest worldwide prevalence and incidence. Estimations from the International Diabetes Federation (IDF) indicate that this number will increase to 665 million within the next 25 years. The main proposed reason for the increasing numbers is the “Western Lifestyle”, characterized by inactivity and high calorie intake. Especially at risk are elderly, where the prevalence of T2DM increases up to 20% [1]. T2DM is associated with multiple complications such as altered micro- and macro-vasculature, nephro- and neuropathies, an overall inflammatory state as well as alterations in bone homeostasis. There is evidence that pre-diabetics already have an altered bone quality and an increased fracture risk [2]. Associated diabetic bone diseases rapidly develop, e.g., diabetic foot syndrome or a Charcot-osteoarthropathy, which affects almost 20% of all diabetics within the first 5 years of the disease [3], often (~20%) resulting in an amputation of the affected limb [4]. Interestingly, T2DM patients often have an increased bone mineral density (BMD) [5,6] with a significantly increased risk for fractures. Furthermore, bone of T2DM patients is characterized by increased amounts of adipose tissue in the bone, altered collagen structure and reduced vascularization (reviewed in [7]). However, in case of a fracture, these patients suffer from impaired fracture healing rich in complications [8,9]. Thus, it was not surprising that a large British study with almost 3000 patients identified diabetes as one of the main risk factors for delayed fracture healing and fracture non-unions [10]. The treatment of bone diseases accounts for a large proportion of the long-term therapeutic costs for diabetics [11], highlighting the need for new therapeutic options.

Fracture healing and wound healing both follow certain basal mechanisms, including the initial local inflammation and granulation, extracellular matrix deposition, and finally the remodeling. Therefore, strategies successfully used for wound healing could be taken into account for fracture healing.

A few years ago, it has been shown that factors secreted by freshly isolated and γ-irradiated peripheral blood mononuclear cells (PBMCs) improved wound healing both in vitro and in a rodent model. The positive effects on keratinocytes, fibroblasts, and endothelial cells have been associated with activation of different MAPKinase signaling pathways [12]. One major cell type involved in fracture healing processes are mesenchymal stem cells (MSCs) [13]. MSCs differentiate into bone cells, chondrocytes, and adipocytes, and are recruited by the inflammatory conditions of the fracture hematoma [14]. It has to be clarified whether the observed effect of the PBMCs on skin cells can be translated to MSCs and how this is affected by pre-/diabetic conditions. Just recently, the data on PBMCs effects on wound healing have been confirmed in a rodent gastric flap and diabetic wound healing model [15,16], as well as a porcine burn model [17]. These follow-up studies came to the conclusion that the secretome of apoptotic PBMCs better supports wound healing than the secretome of unstressed PBMCs, which emphasizes that the activation status of the isolated PBMCs may be crucial. In vitro, the differentiation status of monocytes was shown to be pivotal for appropriate differentiation of MSCs into osteoblasts [18].

Upon vessel rupture, oxygen and pH drop locally at the site of fracture. The resulting hypoxic environment stimulates expression and secretion of chemokines and pro-inflammatory cytokines in the resident bone cells, which in turn attract immune cells to the fracture gap. The immune cells then secrete factors to initiate the healing process. So far it is yet unknown, how pre-/diabetic conditions affect the expression profile of these bone resident cells. However it has been described, that migration of mononuclear cells is delayed under hyperglycemic conditions in vitro [19]. This phenomenon was linked to an accumulation of advanced glycation end products (AGEs) in the hyperglycemic conditions [19]. It is likely that in diabetics, with higher levels of AGEs, fewer mononuclear cells arrive at the site of fracture as compared to corresponding controls. Therefore, application of PBMCs or their secretome should occur during the initial treatment of the fracture, when the early pro-inflammatory phase is expected to happen. This pro-inflammatory state shall be simulated by the applied cells or factors.

However, diabetics are often in a chronic inflammatory state, associated with an altered secretion of cytokines and growth factors by the immune cells—for example the composition of factors found circulating in the serum of pre-/diabetics was shown to inhibit osteogenic differentiation in a transforming growth factor beta (TGF-β) dependent manner [20]. The described PBMC secretome contains TGF-β, which experimentally not only induced migration and proliferation of osteoprogenitor cells [21], but also induced osteoclastogenesis [22], which is expected to be found at later time-points of fracture healing. This needs to be taken into account when considering this treatment option for fracture healing.

The aim of this study was therefore to obtain a deeper insight into how mononuclear cells induce migration of osteoprogenitor cells, i.a. which factors are secreted and which signaling pathways get activated, and how this is affected by pre-/diabetic conditions. Furthermore, to investigate how pre-/diabetic conditions alter the expression and secretion of chemokines in bone resident osteoprogenitor cells, and how this affects migration of mononuclear cells. Overall, this should give hints, whether PBMCs or their secretome can be utilized to support fracture healing in diabetics.

## 2. Results

### 2.1. THP-1 Cells Secreted Factors Induced Migration of Osteoprogenitor Cells

As previously published, THP-1 cell conditioned medium favors migration of osteoprogenitor cells and SCP-1 cells—an effect partly attributed to the secreted TGF-β [21]. With the intention to identify other possible regulatory factors, the THP-1 cell-conditioned medium was fractioned by the molecular weight (MW in kDa) of the secreted factors. Migration of osteoprogenitor cells was measured in the presence of the fractioned THP-1 cell conditioned medium. As expected, the unfractionated THP-1 cell conditioned medium significantly induced (70%) migration of osteoprogenitor cells. This effect was significantly reduced (17% to 35%) when osteoprogenitor cells were stimulated with the fractionated THP-1 cell conditioned media (Figure 1A). Looking closer at the fractionated THP-1 cell conditioned media, the ones with the lower MW had the strongest effect on the osteoprogenitor cells. To investigate which intracellular signaling pathway could be involved in the positive effect THP-1 cell conditioned medium has on migration of osteoprogenitor cells a PathScan^®^ Intracellular Signaling Array test (Cell Signaling Technology, Danvers, MA, USA) was performed with lysates of the osteoprogenitor cells. THP-1 cell conditioned medium strongly induced phosphorylation of P70 S6, P53, GSK-3β, ERK1/2, and p38. At the same time phosphorylation of AMPKα and mTOR was suppressed. In presence of the fractionated THP-1 cell conditioned media these effects were overall reduced for GSK-3β and p38 and even abolished for ERK1/2 and mTOR. Additionally, phosphorylation of SAPK/JNK, Akt, Hsp 27, Stat-1 and Stat-3 was reduced in presence of the fractionated THP-1 cell conditioned media (Figure 1B).

### 2.2. Activation of ERK1/2 Signaling Was Required for the Positive Effect THP-1 Cell Conditioned Medium Has on Migration of Osteoprogenitor Cells

THP-1 cell-conditioned medium activated ERK1/2 signaling in osteoprogenitor cells for at least 4 h. In contrast, the activation of p38 signaling faded already after 1 h (Figure 2A). To investigate the relevance of these two signaling pathways for THP-1 cell-favored migration of osteoprogenitor cells, migration assays were repeated in presence or absence of the p38 inhibitor SB203580 or the ERK1/2 inhibitor U0126. Inhibitors were used in a concentration that was non-toxic to the cells and effectively inhibited phosphorylation of p38 and ERK1/2, respectively (Figure 2B). Inhibition of p38 signaling did not reduce the positive effect THP-1 cell conditioned medium has on osteoprogenitor cells migration. In contrast, inhibition of ERK1/2 signaling almost completely abolished THP-1 cell induced migration of osteoprogenitor cells (Figure 2C).

### 2.3. Pre-/Diabetic Conditions Affected Recruitment of Mononuclear Cells by Osteoprogenitor Cells

In case of a wound or fracture, local hypoxia stimulates expression of chemokines in resident cells that attract immune cells to the site of injury [23]. In vitro hypoxia stimulated osteoprogenitor cells, e.g., the immortalized SCP-1 cells, to secrete chemokines that attract mononuclear cells, e.g., THP-1 cells. Under hyperinsulinemic and hyperglycemic conditions the composition of the secreted chemokines was altered, which inhibited THP-1 cell migration. The effect was more pronounced in hyperglycemic conditions (Figure 3A). To narrow down possible mediators for this effect, the influence of pre-/diabetic conditions and/or hypoxia on gene expression in SCP-1 cells was investigated. Targets were cytokines and chemokines, which are known to be secreted by osteoprogenitor cells and to act on mononuclear cells [24,25], e.g., *M-CSF*, *GM-CSF*, *CXCL9*, *CCL2*, *CCL5*, *CCL7*, *CCL8*, and *CCL13*. Hypoxia strongly induced expression of *M-CSF* and *CCL8* in SCP-1 cells, followed by *GM-CSF*, *CXCL9*, *CCL13*, and *CCL2*. The contrary was observed for expression of *CCL7*. Expression of *CCL5* was not affected by hypoxia in SCP-1 cells. Hyperinsulinemic conditions increased basal expression of *M-CSF*, *CCL13*, and *CXCL9*, and decreased basal expression of *CCL2*, *CCL5*, and *CCL8*. Under hypoxic conditions, hyperinsulinemia increased expression only of *CXCL9* and *CCL7*, but suppressed expression of *M-CSF*, *GM-CSF*, *CCL2*, *CCL5*, *CCL8*, and *CCL13*. Hyperglycemia in turn, increased basal expression of *CCL13*, *CCL7*, and *CXCL9*, and decreased basal expression of *M-CSF*, *CCL2*, *CCL5*, and *CCL8*. Under hypoxic conditions, hyperglycemia had similar effects than hyperinsulinemia, except for the expression of *GM-CSF* and *CCL13*, which were not affected by hyperglycemia (Figure 3B).

### 2.4. Hyperglycemia Affected Chemoattraction of Mononuclear Cells by MCP-1 and MCP-2

As hypoxia failed to increase gene expression of *CCL2* (encoding MCP-1) and *CCL8* (encoding MCP-2) under pre-/diabetic conditions (Figure 3B), corresponding chemokines levels were determined in serum of newly diagnosed type 2 diabetics and respective controls. Both MCP-1 and MCP-2 serum levels were significantly reduced in diabetics, when compared to the controls (Figure 4A).

As expected, migration of THP-1 cells was dose-dependently induced by recombinant human MCP-1 and MCP-2. However, under hyperglycemic conditions, migration of THP-1 cells was significantly suppressed in the presence of MCP-1 or MCP-2 (Figure 4B,C).

### 2.5. Pre-/Diabetic Conditions Altered the Secretome of THP-1 Cells

Our results showed that THP-1 cell conditioned medium favors migration of osteoprogenitor cells by activation of ERK1/2 signaling. ERK1/2 signaling is known to be activated by many cytokines and growth factors. To determine the effect of pre-/diabetic conditions on the secretion of cytokines and growth factors by THP-1 cells, antibody based arrays, namely the human Cytokine Array C5 and the human Growth Factor Array test (RayBiotech, Peachtree Corners, GA, USA), were performed using THP-1 cell conditioned media. Medium conditioned by PBMCs from type 2 diabetics and matched controls were used as reference. The data are summarized as heat map in Figure 5.

Pre-/diabetic conditions, especially hyperglycemia, altered levels of cytokines secreted from THP-1 cells. Many cytokines associated with a pro-inflammatory response, e.g., IL-1, TNF-α, IFN-γ, and IFN-γ responsive CXCL-chemokines, as well as growth factors, e.g., FGFs, PDGF-BB, or IGF-1, were increased in hyperglycemic (HG) THP-1 cell conditioned medium. This is similar to the results observed in media conditioned by PBMCs from diabetics and corresponding controls. Interestingly, under diabetic conditions macrophage migration inhibitory factor (MIF) is increased in media conditioned by THP-1 cells and PBMCs. The inverse trend of reduced levels under diabetic conditions is mainly observed for VEGF-A, MCP-2, IL-6, and M-CSF. Additionally, THP-1 cell conditioned media from pre-/diabetic conditions showed reduced levels of cytokines associated with an anti-inflammatory response, e.g., IL-5, IL-10, and TGF-βs.

### 2.6. THP-1 Cells Failed to Stimulate SCP-1 Cells Migration under Pre-/Diabetic Conditions

THP-1 cell conditioned medium was shown to favor migration of osteoprogenitor cells, e.g., SCP-1 cells. However, it is not clear how the altered cytokine and growth factor levels in the media conditioned by THP-1 cell under pre-/diabetic conditions affect migration of osteoprogenitor cells. Therefore, migration assays were performed (Figure 6A,B). For the assays, SCP-1 cells were stimulated with media conditioned by THP-1 cells under pre-/diabetic conditions. After 2 days, hyperinsulinemia only slightly reduced the area covered by SCP-1 cells in the migration zone. At the same time, hyperglycemia significantly reduced the area covered by SCP-1 cells in the migration zone. To determine, whether the observed effect correlates with cell numbers, the total protein content of the attached cells was determined by SRB staining. The amount of cells showed a slight increase under hyperinsulinemic conditions and significant reduction under hyperglycemic conditions, when compared to control conditions (Figure 6C). A similar result was observed for activation of ERK1/2 signaling. As expected, levels of phosphorylated ERK1/2 were increased in SCP-1 cells exposed to THP-1 cell conditioned medium for 30 min. When compared to control conditions, ERK1/2 phosphorylation was significantly decreased in SCP-1 cells exposed to hyperglycemic THP-1 cell conditioned medium (Figure 6D).

## 3. Discussion

Recently, it has been shown that diabetic conditions reduced bone turnover in a 3D bone co-culture model [26], possibly explaining the reduced stability of bones frequently observed in T2DM patients. The reduced stability of bones increases the fracture risk of T2DM patients [6]. In case of a fracture, the healing process is often accompanied by complications [8,9], such that T2DM is recognized as one of the major risk factors for delayed or impaired bone healing [10]. Steadily increasing numbers of diabetics, emphasize the need for new therapeutic options to support bone regeneration. Different treatment strategies have been developed and tested, including synthetic bone graft substitutes, biologically active substances and stem cells, however, with limited success [27]. In recent years, the immune system attracted notice to research as a key regulator of the bone healing processes. Utilizing these immune cell-mediated mechanisms to support the healing process appears to be intriguing. Already in 2013, it has been shown that factors secreted by PBMCs improved wound healing both in vitro and in a rodent model [12]. Since then, other studies confirmed the positive effect of PBMC secreted factors on healing processes in diverse disease models, including spinal cord injury [28], burns [17], acute myocardial infection [29], or gastric flaps [15]. However, little is known how diabetic conditions affect these mechanisms. Therefore, possible diabetes-dependent alterations in the interplay between osteoprogenitor cells and immune cells have been investigated in this study.

Our data show that pre-/diabetic conditions altered the secretome of osteoprogenitor cells under hypoxia, a condition expected at the site of fracture. This includes chemokines, e.g., MCP-1 and MCP-2, well known to attract monocytic cells [30]. MCP-1 and MCP-2 are frequently found in the fracture gap [24]. Reduced expression and secretion of these two chemokines in response to hypoxia under pre-/diabetic conditions suggests that fewer immune cells are recruited to the fracture hematoma in diabetics. A condition amplified by the finding that hyperglycemia decelerated migration of monocytic cells. A possible explanation might give the increased secretion of MIF by both PBMCs from diabetics and THP-1 cells under diabetic conditions [31]. MIF is a proinflammatory cytokine well known to inhibit the migration of macrophages [30]. However, MIF is not the only cytokine, which secretion is affected in PBMCs from T2DM patients. When compared to metabolically healthy controls, T2DM patients show altered levels of inflammatory cytokines circulating in the blood [32], associated with a chronic inflammatory state. It has been shown that circulating levels of active TGF-β are elevated in T2DM patients, which inhibits maturation of osteoprogenitor cells when exposed over a longer period of time [20,33]. In contrast, a short elevation of active TGF-β is required to attract osteoprogenitor cells to the site of fracture and induce their proliferation early in fracture healing. Inducing an inflammatory response in monocytic THP-1 cells by phorbol 12-myristate 13-acetate further enhanced the positive effect of the THP-1 cell conditioned medium on migration of osteoprogenitor cells [21], suggesting that a chronic inflammation expected in diabetics may not be a limiting factor in this approach. However, our results showed that migration of SCP-1 cells was significantly delayed when exposed to medium condition by THP-1 cells under hyperglycemic conditions. Fibroblasts isolated from diabetic *db/db* mice, showed a similar reduction in migration capacity and limited cellular response to hypoxia [34].

This is in line with our finding: stimulation of osteoprogenitor cells with the secretome of PBMCs, which contains also active TGF-β, induced the cells migration. Activation of TGF-β (44 kDa) requires proteolytic cleavage to obtain the smaller active form of TGF-β (13 kDa). Thus, a positive effect on the migration of osteoprogenitor cells was expected by stimulation with the smaller MW-fraction (10–30 kDa) of the THP-1 cell conditioned medium. However, after fractionation the positive effect on the migration of osteoprogenitor cells was abandoned, suggesting, that a combination of factors is required for the observed positive effects on osteoprogenitor cells. One of these factors might be M-CSF, which is known as a potent inducer of bone healing and remodeling [35]. M-CSF, which should be found in the 50–100 kDa MW-fraction, is induced by hypoxia but not under pre-/diabetic conditions. In contrast, TNF-α (17 kDa) was strongly induced in PBMCs from T2DM patients. TNF-α was shown to affect wound healing by counteracting healing processes in keratinocytes and inducing polarization of macrophages to M1 phenotype [36]. A generally stronger polarization of monocytes into macrophages (both M1 and M2 type) with prolonged inflammation was reported in diabetic mice with impaired tendon healing, leading to increased scar formation [37].

In the study showing that PBMC secretome enhanced wound healing, a cell type dependent activation of MAPKinases was observed [12]. Thus, we determined a cell signaling profile in SCP-1 cells stimulated with THP-1 cell conditioned medium and its MW-fractions. Phosphorylation of P70 S6 and P53 was strongly elevated in SCP-1 cells upon stimulation with THP-1 cell conditioned medium. This effect was still present after fractionation. In contrast, phosphorylation of GSK3β, ERK1/2 and p38, also induced in SCP-1 cells exposed to THP-1 cell conditioned medium, was reduced (GSK3β and p38) or even abolished (ERK1/2) in SCP-1 cells exposed to the fractionated THP-1 cell conditioned media. Chemical inhibition of the ERK1/2 signaling pathway with U0126 significantly reduced the positive effect of THP-1 cell conditioned medium on SCP-1 cell migration. This is in line with the observations made in keratinocytes and fibroblasts during wound healing [12]. In keratinocytes, high glucose levels reduced activation of ERK1/2 signaling and thus migration, an effect that could be restored by addition of MSC-conditioned medium [38]. Interestingly, in our experimental setup the secretome of the THP-1 cells cultured under diabetic conditions failed to induce ERK1/2 signaling in the immortalized MSCs, suggesting that the diabetic conditions affect the communication between osteogenic and monocytic cells. Moreover, our findings are supported by the work with MC3T3 cells, an osteogenic mouse cell line, which observed reduced cell migration and phosphorylation of ERK1/2 and Akt under diabetic conditions [39]. Oncostatin M dependent activation of Stat-3, which was only mildly induced in our SCP-1 cells exposed to the THP-1 cell conditioned medium, was reported to be a key regulator for osteogenic differentiation [40]. Interestingly, activation of Stat-3 was even suppressed in SCP-1 cells exposed to the fractioned THP-1 cell conditioned media, suggesting, that oncostatin M (22–26 kDa) cannot be the sole regulator for the observed positive effect PBMC secreteome has on osteoprogenitor cells [40].

In follow up studies additional activation of Akt was identified as a possible regulatory mechanisms affecting fibroblasts [12], cardiomyocytes [29], dendritic cells and T-cells [41]. In the latter study, lipids were proposed as mediators in the PBMC secretome acting on the immune cells [41]. As Akt signaling was not activated by THP-1 cell conditioned medium in osteoprogenitor cells, this possibility was not further investigated in our setting. The cell specific response is underlined by the work showing that PBMC secretome and to a lesser extent only the secretome from monocytes and not from other single cell types (NK-cells, T-cells) could induce angiogenesis and tube formation by endothelial cells [42]. This work suggests that an additional purification of single cell types is not beneficial to the proposed attempt to use cell secretomes to improve bone healing.

In epithelial cells reduced migration by diabetic conditions was associated with induction of reactive oxygen species (ROS) [43]. Increased levels of ROS are associated with an increased inflammatory response, and thus expression and secretion of MCP-1. Interestingly, MCP-1 was observed to be elevated only in prediabetic obese but not freshly diagnosed diabetics when a chronic inflammatory state is expected [20]. MCP-1 is reported to stimulate not only migration of mononuclear cells, but also of osteoprogenitor cells. By regulating precursor cells of bone resorbing osteoclasts and bone forming osteoblasts, MCP-1 is thought to play a crucial role in bone remodeling [44,45,46,47]. As both MCP-1 and MCP-2 were basically reduced in T2DM patients’ serum, this could contribute to delayed fracture healing in these patients [25]. Interestingly, migration of leucocytes was even accelerated in mouse models of type 1 and type 2 diabetes, however, only when considering tissue infiltration in basal and stimulated conditions. The migrating cells itself showed impaired function [48], which is in line with our observations. A possible explanation is given by the work showing that migration through collagen IV was increased in diabetic monocytes, but their reaction to oxidized collagen was reduced [49]. A similar behavior was found in a type 1 diabetes model where the monocytes were less strongly attracted by MCP-1 (*CCL2*) and MIP-1α (*CCL3*) but stronger to MIP-3β (*CCL19*). Furthermore, monocytes obtained from diabetics showed stronger adherence to fibronectin than the ones obtained from non-diabetic controls [50]. This is in line with another in vitro study, which showed that moderate hyperglycemia induced adhesion and trans-endothelial migration of monocytes [51].

These complex data show that monocytic cells are affected by pre-/diabetic conditions but depending on the environment change their behavior. This suggests, that simply adding autologous PBMCs in wounds or fracture gaps of diabetics will most likely not support the healing process, as their function is already altered. However, this might be affected by the individual medication as another study with HUVECs (human umbilical vein endothelial cells) suggests. In line with other studies, migration and tube formation by HUVECs was stimulated by macrophages. However, these effects were abolished in HUVECs obtained from type 2 diabetic mice but were rescued by treatment with the anti-diabetic drug sitagliptin. In vivo, the same study showed an altered M1/M2 polarization around bone-implanted titanium screws followed by reduced bone formation around the screws and reduced angiogenesis in the diabetic mice [52]. These data suggest that alterations of monocytes in diabetics could also negatively influence other cell types involved in fracture healing like the here mentioned endothelial cells but also chondrocytes [53] or osteoclasts [54]. However, further studies must prove these complex interactions, e.g., in a diabetic mouse model. For example, a study proposed that bone marrow MSCs derived from diabetic mice are more efficient for adenoviral BMP2 therapy to induce bone regeneration than bone marrow MSCs from control mice [55].

A regulatory function of certain anti-diabetic drugs could be already shown for the levels of active TGF-β in serum of T2DM patients [56]. Thus, adaption of the medication could help to lower the effects of diabetes on bone metabolic changes, however in case of a fracture a timely treatment is required. As autologous transplantation of PBMCs has its limitations, applying the secretome of healthy PBMCs seems to be a promising factor to enhance migration and differentiation of osteogenic cells and to support a pro-inflammatory phenotype of immune cells present in the fracture gap. Two studies in mice support our findings of a dysregulation of the fracture gap environment: application of progranulin induced a shift of TNFR1-dependent TNF-α signaling to TNFR2-dependent one and improved fracture healing in T2DM mice [57]. A second study could show altered inflammatory conditions in the fracture callus of T2DM mice and an altered recruiting of MSCs from the stem cell niche. The delayed healing could be rescued by activation of Indian Hedgehog signaling followed by activation of ERK1/2 and Akt signaling [58]. This is in line with our results which showed a crucial role for ERK1/2 in vitro. Altered signaling and inflammatory conditions could also be confirmed in femoral head samples from human diabetic patients in that study [58].

As our data also show diabetes-dependent alterations in the secretome of osteoprogenitor cells, it is feasible that a combination of PBMC and MSC conditioned media is even more effective in boosting fracture healing in diabetics [59,60,61,62,63]. Likewise, specific challenging of the cells, e.g., γ-irradiation, apoptotic or inflammatory stimuli [42,64], may further customize the expected response for specific cell types. Noteworthy, the way of application of the secretome may be relevant. Factors have to be concentrated for local application, e.g., obtained by lyophilization, where the functionality of the factors should be preserved [12]. A study with a diabetic mouse model (e.g., *db/db* mouse) could be used to further prove the hypothesis for the role of PBMCs in diabetic fracture healing and should be used to confirm the here shown data in a more complex approach.

## 4. Materials and Methods

Chemicals, reagents, culture medium, and its supplements were obtained from Sigma-Aldrich/Merck (Darmstadt, Germany) or Carl Roth (Karlsruhe, Germany).

### 4.1. Patient Material

All experiments with human materials were performed in accordance with the Declaration of Helsinki (1964) in its latest amendment. PBMCs were isolated from venous blood and primary human osteoprogenitor cells were isolated from femoral heads or tibia plateaus of patients that received total joint replacement at the BG Trauma Center Tübingen. Patient material was collected with written consent of the patients and did not alter the surgical procedure. Tissues from (potential) tumor patients, patients with viral or bacterial infections and patients unable to give their consent were excluded from this study.

#### 4.1.1. PBMCs

PBMCs, isolated from freshly taken EDTA blood, were used as reference for the experiments. Briefly, venous blood was collected with a butterfly needle into 9 mL tubes (S-Monovette 9 mL, Sarstedt, Sarstedt, Germany). In total 7.5 mL blood were layered onto 5 mL of lymphocyte separation medium (PAA Laboratories, Pasching, Austria) and centrifuged for 20 min at 1000× *g* without breaks. The PBMC layer was transferred into a fresh tube and washed twice with PBS. The cells were counted and seeded in a density of 5 × 10^5^ cells/mL. For experiments the PBMCs were seeded in a density of 5 × 10^5^ cells/mL and cultured at 37 °C (5% CO_2_, humidified atmosphere).

#### 4.1.2. Osteoprogenitor Cells

Osteoprogenitor cells, which served as reference for this study, were isolated by collagenase II digestion as published [25]. Briefly, mechanically minced spongy bone was washed with PBS to remove blood. Cleared pieces of spongy bone were immersed in sterile 0.7% collagenase II solution and incubated at 37 °C for 1–2 h. Released osteoprogenitor cells were removed from the bone pieces by two sequential washing steps with PBS. After centrifugation (1000× *g* for 10 min) cells were resuspended in culture medium (MEM/Ham’s F12, 5% FCS, 100 U/mL penicillin, 100 μg/mL streptomycin, 50 μM L-ascorbate-2-phosphate, 50 μM β-glycerol-phosphate). Cells were expanded until passage 2–3 before being used for experiments. For osteogenic differentiation, the basic medium was supplemented with 1% FCS, 200 μM L-ascorbate-2-phosphate, 5 mM β-glycerol-phosphate, 25 mM HEPES, 1.5 mM CaCl_2_, and 100 nM dexamethasone.

### 4.2. Cell Lines

#### 4.2.1. THP-1 Cells

THP-1 cells (DSMZ-German Collection of Microorganisms and Cell cultures GmbH, Braunschweig, Germany) were cultured in RPMI 1640 Medium with 5% FCS (37 °C, 5% CO_2_, humidified atmosphere). Medium was changed twice a week. Cell density was kept between 5 × 10^5^ and 2 × 10^6^ cells/mL. For experiments the THP-1 cells were seeded in a density of 5 × 10^5^ cells/mL and cultured at 37 °C (5% CO_2_, humidified atmosphere).

#### 4.2.2. SCP-1 Cells

The immortalized bone marrow-derived mesenchymal stem cell line SCP-1 was kindly provided by Prof. Matthias Schieker [65]. SCP-1 cells were cultured (37 °C, 5% CO_2_, humidified atmosphere) in MEMα medium supplemented with 5% FCS. Medium was changed twice a week with sub-culturing at 80–90% confluence. Cells were differentiated in basic medium supplemented with 1% FCS, 200 μM L-ascorbate-2-phosphate, 5 mM β-glycerol-phosphate, 25 mM HEPES, 1.5 mM CaCl_2_, and 100 nM dexamethasone.

### 4.3. Cell Culture Conditions

#### 4.3.1. Simulation of Pre-/Diabetic Conditions

To simulate the development of type 2 diabetes mellitus in vitro, glucose and insulin were supplemented to the medium as described before [33]:
Normoglycemic control conditions (Ctrl): To equalize osmolarity compared to hyperglycemic conditions (HG) mannitol was added to the medium to reach a concentration of 25 mM glucose + mannitol.Hyperinsulinemic “prediabetic” conditions (HI): Ctrl medium was supplemented with 160 I.U./L insulin (Actrapid, NovoNordisk, Bagsværd, Denmark).Hyperglycemic “diabetic” conditions (HG): glucose concentration of the basal medium was increased to 25 mM.

#### 4.3.2. Hypoxia Induction

Hypoxia was induced by increasing the medium height (3-fold), which reduced the partial oxygen pressure [66]. Hypoxia was confirmed with the Image-iT™ Red Hypoxia Reagent (ThermoFisher Scientific, Waltham, MA, USA). After 24 h, CoCl_2_ (100 µM) was added to the culture medium for 2 h to further increase Hif-1α levels.

### 4.4. Cell Migration Assays

#### 4.4.1. Scratch Assay and ORIS™ Migration Assay for Osteoprogenitor Cells

To analyze migration of the adherent osteoprogenitor cells a classical scratch assay, where the confluent cell layer is wounded with a pipette tip, was performed. To obtain a more defined migration zone the ORIS^TM^ migration assay inserts (Platypus, Madison, WI, USA) were used. Briefly, sterile inserts were placed in 96-well plates and osteoprogenitor cells were seeded in a concentration of 2 × 10^5^ cells/well. After cell adherence, inserts were removed and medium was changed to the different THP-1 cell conditioned media (in 50% culture medium). For both assays images were taken immediately (0 h) and after 45 h. Cells were fixed and stained with SRB for better visualization. Migration was determined in percent of the migration zone (cell free area at 0 h) covered by cells after 45 h.

#### 4.4.2. Boyden Chamber Assay for Migration of Mononuclear Cells

For the analysis of migration of non-adherent mononuclear cells a Boyden chamber assay was used. Briefly, cells were incubated with 2 µM Calcein-AM (ATT Bioquest, Sunnyvale, CA, USA) for 30 min to label the viable cells. After washing with PBS, cells (1 × 10^6^ cells) were immediately applied to 8 µm cell culture inserts placed in a 24-well plate. The bottom part of the well contained the chemoattractant—either SCP-1 cell conditioned medium, MCP-1 or MCP-2. THP-1 cells migrated into the bottom part of the well within 6.5 h were quantified fluorometrically (λ_ex_ = 485 nm and λ_em_ = 520 nm) with the omega microplate reader (BMG Labtech, Ortenberg, GER).

### 4.5. Quantification of Protein Levels

#### 4.5.1. PathScan^®^ Intracellular Signaling Array with SCP-1 Cell Lysates

A PathScan^®^ Intracellular Signaling Array (Cell Signaling Technology, Danvers, MA, USA) was used to assess activation of intracellular signaling pathways. For the assay, SCP-1 cells were stimulated for 30 min with THP-1 cell conditioned media before lysing the cells with the provided lysis buffer. The array was performed according to the manufacturer’s instructions. Chemiluminescent signals were detected with the Chemocam (Intas, Göttingen, Germany). Signal intensities, quantified using the ImageJ software (Version 1.52, NIH, Bethesda, MD, USA), were normalized to the internal controls.

#### 4.5.2. Fractionation of THP-1 Cell Conditioned Medium

5 mL of THP-1 cell conditioned medium were serially applied to spin columns with different size cut-offs (Vivaspin 6, Sartorius, Göttingen, Germany). Medium was centrifuged at 4000× *g* for 15 min. For each size cut-off the flow-through was collected and applied to the next smaller size cut-off column. The cut-off sizes resulted in fractions with protein sizes of <3 kDa, 3–5 kDa, 5–10 kDa, 10–30 kDa, 30–50 kDa, 50–100 kDa and >100 kDa. The individual fractions were filled up with plain RPMI medium to compensate for the reduction in volume.

#### 4.5.3. Western Blot with SCP-1 Cell Lysates

SCP-1 cells were stimulated for 30 min with THP-1 cell conditioned media before lysing the cells with ice-cold RIPA buffer (50 mM TRIS, 250 mM NaCl, 2% NP40, 2.5 mM EDTA, 0.1% SDS, 0.5% deoxycholic acid, protease and phosphatase inhibitors, pH = 7.2). 30 µg total protein, quantified by micro-Lowry, was separated by SDS-PAGE and transferred to nitrocellulose membranes. Unspecific binding sites were blocked with 5% BSA in TBST (25 mM TRIS, 137 mM NaCl, 2.7 mM KCl, 0.05% Tween-20, pH = 7.4) for 1 h at RT. After overnight incubation with primary antibodies at +4 °C (anti-phospho-p42/44 (ERK1/2) #4370 and anti-phospho-p38 #5611 from Cell Signaling Technologies diluted 1:1000 in TBST/anti-GAPDH #G9545 from Sigma-Aldrich/Merck diluted 1:5000 in TBST), membranes were incubated with the corresponding goat-anti-rabbit peroxidase-labeled secondary antibody (#sc-2004 from SantaCruz Biotechnology (Heidelberg, Germany) 1:10,000 in TBST) for 2 h at RT. After washing with TBST membranes were covered with chemiluminescent substrate solution (1.25 mM luminol, 0.2 mM p-coumaric acid, 0.03% H_2_O_2_ in 100 mM TRIS, pH = 8.5) and chemiluminescent signals were immediately detected with the Chemocam. Signals were quantified with the ImageJ software.

#### 4.5.4. Human Cytokine Array C5 and Human Growth Factor Array with Media Conditioned by Mononuclear Cells

The human Cytokine Array C5 and the human Growth Factor Array (RayBiotech, Peachtree Corners, GA, USA) were used to characterize media conditioned by mononuclear cells. For the assay, THP-1 cells or PBMCs (5 × 10^5^ cells/mL) were cultured for 48 h at 37 °C (5% CO_2_, humidified atmosphere). Conditioned media were obtained after centrifugation at 1000× *g* for 10 min. The array was performed according to the manufacturer’s instructions. Chemiluminescent signals were detected with the Chemocam. Signal intensities, quantified using the ImageJ software, were normalized to the internal controls.

#### 4.5.5. Enzyme-Linked Immunosorbent Assay (ELISA) with Serum Samples

Levels of MCP-1 and MCP-2 in patients’ sera were quantified with the help of ELISA kits (#900-K31 and #900-K41, Peprotech, Hamburg, Germany). The ELISA was performed as indicated by the manufacturer using a 1:5 dilution of the serum samples. ABTS signals were detected photometrically (λ = 405 nm–650 nm) with the omega microplate reader. Non-linear curve fit for the standard curve, interpolation, and transformation of data was done with the GraphPad Prism software version 8 (GraphPad, San Diego, CA, USA).

### 4.6. RNA Isolation and RT-PCR

For mRNA isolation, cells were lyzed with guanidinium thiocyanate-phenol solution. Total mRNA was isolated by phenol-chloroform extraction. The mRNA pellet was washed twice with 70% ethanol and was then resuspended in DEPC water. Total mRNA content was measured photometrically (λ = 260 nm, 280 nm and 320 nm) with the omega microplate reader. After successful integrity check by electrophoresis, the mRNA was converted into cDNA using the first strand cDNA synthesis kit (ThermoFisher Scientific) according to the manufacturers’ instructions. RT-PCR was carried out with the 2× Red Taq Mastermix (Biozym, Oldendorf, Germany). Optimized PCR conditions for each primer set are given in Table 1. Primers were designed with the help of primer blast with the respective gene bank accession number listed in the table.

### 4.7. Sulforhodamine B (SRB) Staining

To quantify cells by total protein content, adherent cells were fixed with ice-cold 99% EtOH for at least 1 h. Plates were washed once with tap water and air-dried. Then, cells were covered with SRB solution (0.4% SRB in 1% acetic acid) for 30 min. Unbound SRB was washed off the cells with 1% acetic acid. Bound SRB was resolved with 10 mM unbuffered Tris solution and quantified photometrically (λ = 565 nm–690 nm).

### 4.8. Statistical Analysis

Results are presented as bar diagrams (mean ± 95% C.I.) or box plots (Tukey’s to visualize outliers). Each experiment was repeated at least 3 times (N ≥ 3) with a minimum of two independent replicates (*n* ≥ 2). The exact number of biological (N) and technical replicates (*n*) for each experiment is given in the figure legends. Statistical analyses were performed using the GraphPad Prism Software version 8 (San Diego, CA, USA). Depending on the experimental setup, data sets were compared by non-parametric Kruskal-Wallis test, followed by Dunn’s multiple comparison test, or non-parametric 2-way ANOVA, followed by Sidak’s multiple comparison test. A *p*-value below 0.05 was considered statistically significant.

## 5. Conclusions

In the present work, we analyzed the role of PBMCs for the migration of osteoprogenitor cells and SCP-1 cells with regard to diabetic conditions. Our data clearly show that pre-/diabetic conditions altered the secretome of SCP-1 cells under hypoxia, which in turn affected THP-1 cell migration. The secretome of these monocytic cells strongly induced osteoprogenitor cell migration, in a ERK1/2-dependant manner. This effect is abolished under pre-/diabetic conditions, especially under hyperglycemia.

Our data suggest that under pre-/diabetic conditions fewer monocytic cells are recruited to the site of fracture and that their secretome is altered. This in turn affects migration of osteoprogenitor cells. In summary, this may contribute to the delayed fracture healing often observed in diabetics. Thus, restoration of the secreted factors in diabetics could probably improve diabetic fracture healing. Animal studies have proposed that autologous transplantation of PBMCs or their secretome improve the healing process. However, our data challenge this approach for diabetics, as the secretome of these cells is altered. In diabetics the application of a “normal” PBMC secretome, especially when it can be functionally provided as lyophilisate for immediate use, might be a suitable alternative. However, further in vivo studies are required to provide evidence.

## Figures and Tables

**Figure 1 ijms-22-09212-f001:**
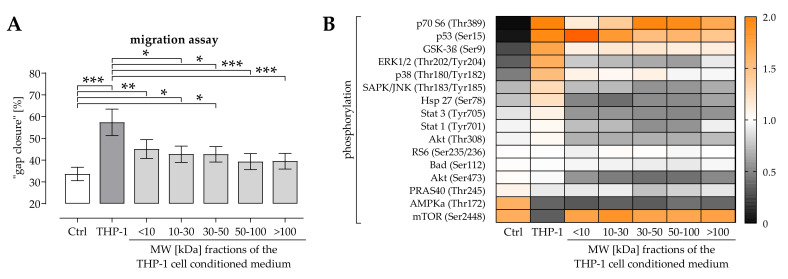
THP-1 cell conditioned medium induces MAPKinase signaling in and cell migration of osteoprogenitor cells. THP-1 cells were cultured for 45 h to obtain THP-1 cell conditioned medium. THP-1 cell conditioned medium was fractioned by molecular weight (MW) with spin columns (Sartorius, Göttingen, GER). (**A**) Migration of osteoprogenitor cells was determined with a scratch assay. Data are presented as bar chart (mean ± 95% C.I.). N = 5 and *n* = 6. Data were compared by non-parametric Kruskal-Wallis test, followed by Dunn’s multiple comparison test. * *p* < 0.05, ** *p* < 0.01 and *** *p* < 0.0001 as indicated. (**B**) Pools of the corresponding osteoprogenitor cell lysates (30 min stimulation with THP-1 cell conditioned medium) were examined with a PathScan^®^ Intracellular Signaling Array test to determine activation of intracellular signaling pathways. Mean data are presented as heat map. Ctrl = control.

**Figure 2 ijms-22-09212-f002:**
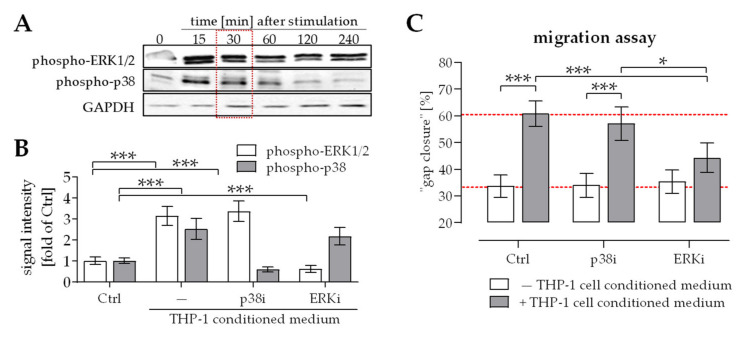
ERK1/2 activation is required for THP-1 cell-favored migration of osteoprogenitor cells. (**A**) Representative Western blot images showing ERK1/2 and p38 phosphorylation in osteoprogenitor cells 0, 15, 30, 60, 120, and 240 min after stimulation with THP-1 cell conditioned medium. (**B**) ImageJ was used to quantify ERK1/2 and p38 phosphorylation (PathScan^®^ Intracellular Signaling Array) in SCP-1 cells 30 min after stimulation with THP-1 cell conditioned medium in the absence (Ctrl) or presence of p38 inhibitor (p38i) SB203580 or ERK1/2 inhibitor (ERKi) U0126. (**C**) Migration of osteoprogenitor cells was determined using a scratch assay. Data are presented as bar chart (mean ± 95% C.I.). N = 3 and *n* = 3. Data were compared by non-parametric 2-way ANOVA, followed by Sidak’s multiple comparison test. * *p* < 0.05 and *** *p* < 0.0001 as indicated.

**Figure 3 ijms-22-09212-f003:**
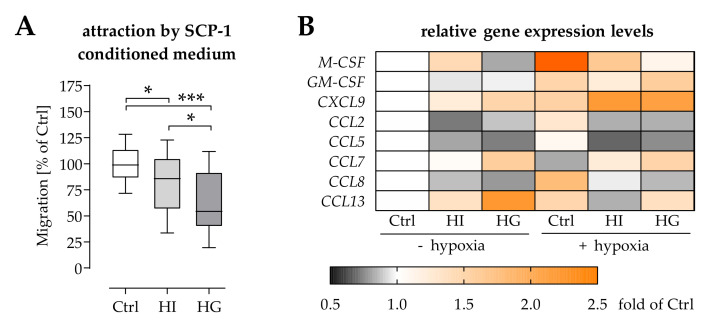
Diabetic conditions affect expression and secretion of factors in SCP-1 cells required to attract THP-1 cells. (**A**) SCP-1 cells were cultured under normoglycemic (Ctrl), hyperinsulinemic (HI), or hyperglycemic (HG) conditions for 48 h to obtain SCP-1 cell conditioned media. Migration of THP-1 cells towards these SCP-1 cell conditioned media was determined by Boyden chamber migration assay. Data are presented as box plot. Data were compared by non-parametric Kruskal-Wallis test, followed by Dunn’s multiple comparison test. * *p* < 0.05 and *** *p* < 0.0001 as indicated. N = 4 and *n* = 4. (**B**) After 24 h in the presence or absence of hypoxia, relative gene expression levels of *M-CSF*, *GM-CSF*, *CXCL9*, *CCL2*, *CCL5*, *CCL7*, *CCL8*, and *CCL13* in the conditioned SCP-1 cells were determined by semiquantitative RT-PCR. Amplicons were seperated twice by electrophoresis, to minimize loading errors. Signal intensities were quantified using the ImageJ software (NIH, Bethesda, MD, USA) and normalized to the control group. Mean data (N = 4, *n* = 2) are presented as heat map.

**Figure 4 ijms-22-09212-f004:**
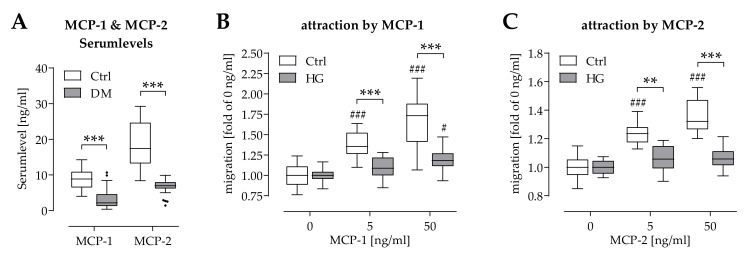
Attraction of THP-1 cells by MCP-1 and MCP-2 is impaired by pre-/diabetic conditions. (**A**) Levels of MCP-1 and MCP-2 in serum of newly diagnosed type 2 diabetics (N = 34) and respective controls (N = 32) were determined by ELISA. (**B**) Migration of THP-1 cells in the presence of MCP-1 or (**C**) MCP-2 under normoglycemic (Ctrl) and hyperglycemic (HG) conditions was determined with the help of Boyden chambers (N = 4 and *n* = 3). Data are presented as box plots. Data were compared by non-parametric 2-way ANOVA, followed by Sidak’s multiple comparison test. ** *p* < 0.01 and *** *p* < 0.0001 as indicated. ^#^
*p* < 0.05 and ^###^
*p* < 0.0001 as compared to the respective control sample without MCPs.

**Figure 5 ijms-22-09212-f005:**
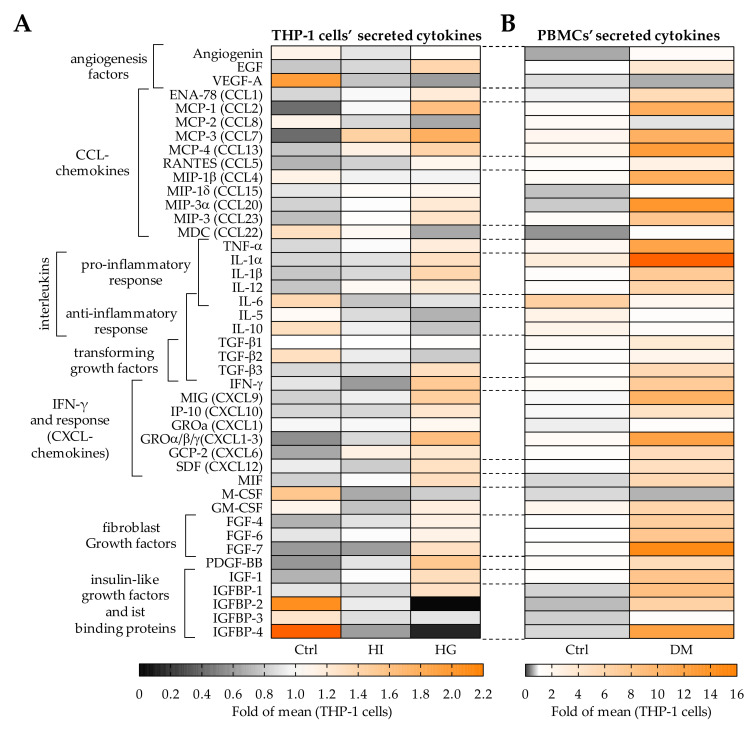
Pre-/diabetic conditions alter the composition of factors secreted by THP-1 cells and PBMCs. (**A**) THP-1 cells were cultured under normoglycemic (Ctrl), hyperinsulinemic (HI), or hyperglycemic (HG) conditions for 48 h to obtain THP-1 cell conditioned medium. (**B**) PBMCs isolated from type 2 diabetics (DM) and respective controls (Ctrl) were cultured for 48 h to obtain PBMC conditioned medium. Relative cytokine levels in the conditioned media (pools of N = 6) were determined using the human Cytokine Array C5 and the human Growth Factor Array. Mean data are presented as heat map.

**Figure 6 ijms-22-09212-f006:**
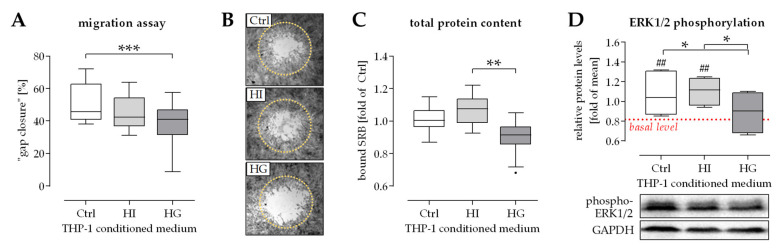
Pre-/diabetic conditions impair THP-1 cells-favored migration of SCP-1 cells. THP-1 cells were cultured under normoglycemic (Ctrl), hyperinsulinemic (HI), or hyperglycemic (HG) conditions for 48 h to obtain THP-1 cell conditioned media. (**A**) Migration of SCP-1 cells in the presence of these THP-1 cell conditioned media was determined using the Oris^TM^ cell migration assay (Platypus Technologies, Madison, WI, USA). (**B**) Representative image of the cell migration assay after 45 h of incubation. The dotted circle marks the initial detection zone (Original magnification 2x) (**C**) Total protein content was determined by SRB staining. (**D**) Levels of phosphorylated ERK1/2 were determined by Western blot in SCP-1 cells exposed to THP-1 cell conditioned media. GAPDH was used for normalization. Signal intensities were quantified using the ImageJ software. Data are presented as box plots. N = 4 and *n* = 4. Data were compared by non-parametric Kruskal-Wallis test, followed by Dunn’s multiple comparison test. * *p* < 0.05; ** *p* < 0.01 and *** *p* < 0.0001 as indicated. ^##^
*p* < 0.01 as compared to the respective control sample without THP-1 conditioned medium (basal level).

**Table 1 ijms-22-09212-t001:** List of primers, their sequences, and the corresponding PCR conditions.

Target	Gene Bank Accession Number	Sequence Forward Primer	Sequence Reverse Primer	T_a_ [°C]	No. of Cycles	Amplicon Size [bp]
*GAPDH*	NM_002046.7	GTCAGTGGTG GACCTGACCT	AGGGGTCTAC ATGGCAACTG	56	25	420
*CCL2*/MCP1	NM_002982.4	CCTTCATTCC CCAAGGGCTC	GGTTTGCTTG TCCAGGTGGT	60	35	236
*CCL5*	NM_002985.3	ATCCTCATTG CTACTGCCCTC	GCCACTGGTTA GAAATACTCC	60	40	135
*CCL7*/MCP3	NM_006273.4	CTTGCTCAGC CAGTTGGGATT	CCACTTCGTG TGGGGTCAG	60	35	183
*CCL8*/MCP2	NM_005623.3	ACTTGCTCAGC CAGATTCAGTT	CCCATCTCTC CTTGGGGTCA	60	35	185
*CCL13*/MCP4	NM_005408.3	CAGCCAGATG CACTCAACGTC	CTCCTTTGGG TCAGCACAGA	60	35	168
*CXCL9*	NM_002416.3	CAGGCTCAAAAT CCAATACAGGAG	TACTGGGGTT CCTTGCACTC	58	40	124
*CSF1*/M-CSF	NM_000757.6	AAGTTTGCCT GGGTCCTCTC	CCACTCCAT CATGTGGCT	60	40	289
*CSF2*/GM-CSF	NM_000758.4	GAGACACTGC TGCTGAGATGA	GAGGGCGTG CTGCTTGTA	64	40	180

## Data Availability

The datasets generated during and/or analyzed during the current study are available from the corresponding author on reasonable request.

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
