# Peer review of "Altered Secretome of Diabetic Monocytes Could Negatively Influence Fracture Healing—An In Vitro Study"

_ijms, 2021, doi:10.3390/ijms22179212_

Round 1

Reviewer 1 Report

The manuscript presents data that mononuclear cells promoted the migration of osteoprogenitor cells and this effect was reduced in diabetic conditions. This may have a potential application in bone fracture healing in diabetics. The authors have reported a good amount of data. However, there are issues in the logic of the data, the objective of the manuscript, the methods chosen, and the language et al.

1 The title is too broad and may confuse the readers that the manuscript is an application-driven project instead of a mechanism-driven project. There was no application reported in the manuscript to heal bone fracture using monocytes and their secretome.

2 The logic of the project in the abstract is so vague. It is hard to tell what is the aim of the project in the abstract.

3 It is hard to link the logic of the title, the abstract and the aim of the project (line 80-) together.

4 The authors used two types and three different migration assays. They did not explain why each assay was chosen. Especially, both the scratch assay and “commercial Oris” assay are for the study of collective migration even there are some differences.

5 Inflammatory cells, mesenchymal progenitor cells, endothelial cells, chondrocytes, osteoblasts, and osteoclasts are all involved in bone fracture healing. Using a single mesenchymal cell line to model all the cells of bone fracture healing is not sufficient or adequate. 

6 The migration assay time was documented as “48 h” in Figure 6 but “45 h” in “Materials and Methods” line 448.

7 line 435 “4.2.2 Hypoxia induction: Hypoxia was induced by increasing the medium height (3-fold)”. The degree of hypoxia or the reduction of oxygen pressure is not in linear proportion to the height of the medium. (see: Wenger, R. H.; Kurtcuoglu, V.; Scholz, C. C.; Marti, H. H.; Hoogewijs, D. Frequently Asked Questions in Hypoxia Research. Hypoxia (Auckl) 20153, 35–43 )

A hypoxia assay without proper specific equipment is very unverifiable and the “Image-iT™” related staining cannot improve the reliability either. 

8 The detailed information for fractioning conditioned medium by spin columns is missing the materials and methods. 

9 Non-standard English usage for scientific writing should be avoided. For example, “CO “is not one of the abbreviations for “control”; “with the help of Boyden chambers” is an odd expression. 

Author Response

We would like to thank for the thorough revision of our manuscript and the oportunity for revision. Please find attached the detailed answers to the questions raised by the reviewer.

Reviewer 2 Report

This is an interesting study the possible role of PBMCs specially the monocytic cells in the migration of osteoprogenitor cells  and their suggest their possible role in the fracture healing in both pre/ and diabetic conditions. The result is interesting, however, to confirm such hypothesis, a parallel study using diabetic mouse model such as Streptozitocin induced diabetic mouse is mandatory with induction of fracture and analysis of the parameters in such model. 

Author Response

We would like to thank for the thorough revision of our manuscript and the oportunity for revision. The reviewer made a good point. We agree that more studies including animal studies are necessary to confirm our hypothesis, but we think that our work opens up a good starting point for further studies. Therefore, we emphasized more the in vitro nature of our study both in the title and abstract and added this point to the discussion and conclusion.

Reviewer 3 Report

Regarding to the submitted manuscripts “Monocytes and their secretome as possible therapeutics to favor bone regeneration in diabetics”. Overall, I find this a good and interesting research topic. But, the current form is not acceptable, there are many issues which should be addressed:

  1. For getting conclusions in manuscript the present data are not enough, I think authors should have the in vivo data from animal models
  2. In the figure legends, what does it means N and n? That makes the reader confusing, you should clearly indicate it somewhere
  3. In the introduction, you should write more about type 2 diabetes such as causes, susceptible subjects, ...
  4. In the Introduction, I think the authors should add some related studies and limitations of those studies, which will help increase the urgency of this study.
  5. It is not currently clear how pre-diabetic conditions affect the expression of bone resident cells. I hope in the future you will have detailed studies on this issue.
  6. You analyzed the role of PBMCs for the migration of osteoprogenitor cells and SCP-1 cells regarding to diabetic conditions. So, is PBMC and SCP-1 related to each other?
  7. I found a study “Effect of Hyperglycemia on Human Monocyte Activation” (Nandy, Janardhanan et al. 2011). Hope it will be useful for your article.
  8. I think “The Impact of Type 2 Diabetes on Bone Fracture Healing” (Carlos Marin 2018) will be a good reference for your study.
  9. In the Results section, lines 90-91 on page 2, the authors mention: "With the intention of identifying other possible regulatory factors, the THP-1 plot adjusted for the medium is segmented according to the molecular weight (MW in kDa) of the secreted elements" the author should explain the reference source and how to adjust the medium in the THP-1 plot?
  10. "THP-1 cell conditioned medium activated ERK1/2 signaling in 120. osteoblasts for at least 4 hours. In contrast, the activation of p38 signaling dims after 1 h.” How this change will affect diabetic patients as well as bone regeneration, let's clarify the molecular biology in this section?
  11. In the Materials and Methods section, the authors need to add experimental methods on mice to increase the objectivity of the research. The author can refer to the research of (Park, Kim et al. 2018).
  12. The sequence of primers in Table 1 is designed by yourself or from previous studies? Please mention the reference.

References:

Carlos Marin, o. (2018). "The Impact of Type 2 Diabetes on Bone Fracture Healing."

Nandy, D., R. Janardhanan, D. Mukhopadhyay and A. Basu (2011). "Effect of hyperglycemia on human monocyte activation." J Investig Med 59(4): 661-667.

Park, S. Y., K. H. Kim, C. H. Park, S. Y. Shin, I. C. Rhyu, Y. M. Lee and Y. J. Seol (2018). "Enhanced Bone Regeneration by Diabetic Cell-Based Adenoviral BMP-2 Gene Therapy in Diabetic Animals." Tissue Eng Part A 24(11-12): 930-942.

Author Response

We would like to thank for the thorough revision of our manuscript and the opportunity for revision. Please find attached the detailed answers to the questions raised by the reviewer.

Round 2

Reviewer 1 Report

The quality of the manuscript has been improved.

Author Response

We would like to thank the reviewer for his/her benevolent estimate of our manuscript.

Reviewer 2 Report

I would like to thank the authors for revising their manuscript. I realize the impact of thier invitro study. However, I recommend the authors to continue their research using invivo studiesd including diabetic animal  models, to confirm  and evaluate the current studies. Therefore, the current manuscript could be accepted in the current form and hope in the future, you could continue invivo study to simulate that of the current invitro study.

Author Response

We would like to thank the reviewer for his/her review. We understand the importance of in vivo studies, therefore, in future this point has to be addressed by our group – especially when attempting a therapy which includes triggering immune cells to support fracture healing. The in vitro assays are considered as required work to reduce and refine the animal experiments based on the 3R principle.

Reviewer 3 Report

The manuscript has been revised, and partially improved. I have expected they will do more experiments to have  in vivo data from animal models, and keep there conclusions 

Author Response

We would like to thank the reviewer for his/her comment. We completely understand the importance of in vivo studies. Unfortunately, including animal experiments in this study was not possible for us. However, in future this point has to be addressed by our group – especially when attempting a therapy which includes triggering immune cells to support fracture healing. The in vitro assays presented here are considered as required work to reduce and refine the animal experiments based on the 3R principle.

Due to the importance of the in vivo studies, we included more details of already performed animal experiments and hope that it can improve the quality of the manuscript.